# Single-Domain Antibodies That Specifically Recognize Intact Capsids of Multiple Foot-and-Mouth Disease Serotype O Strains

**DOI:** 10.3390/vaccines13050500

**Published:** 2025-05-08

**Authors:** Michiel M. Harmsen, Nishi Gupta, Quillan Dijkstra, Sandra van de Water, Marga van Setten, Aldo Dekker

**Affiliations:** Wageningen Bioveterinary Research, P.O. Box 65, 8200 AB Lelystad, The Netherlandsquillan.dijkstra@wur.nl (Q.D.); sandra.vandewater@wur.nl (S.v.d.W.); marga.vansetten@wur.nl (M.v.S.)

**Keywords:** foot-and-mouth disease, vaccine, nanobody, single-domain antibody, antigenicity

## Abstract

Background/Objectives: Intact (146S) foot-and-mouth disease virus (FMDV) particles easily dissociate into 12S particles with a concomitant decreased immunogenicity. Vaccine quality control with 146S-specific single-domain antibodies (VHHs) is hampered by the high strain specificity of most 146S-specific VHHs. This study aimed to isolate 146S-specific VHHs that recognize all serotype O strains. Methods: Biopanning was performed with the FMDV strain O/SKR/7/2010 146S, using a secondary library of mutagenized M170F VHH that did not recognize O/SKR/7/2010 or using phage-display libraries from llamas immunized with other serotype O strains. Novel VHHs were yeast-produced and their strain-, particle-, and antigenic-site specificities were determined by ELISA. Results: M170F mutagenesis did not improve the cross-reaction with O/SKR/7/2010. However, selection from immune libraries resulted in four VHHs that exhibited high 146S specificity for all five serotype O strains analyzed. These VHHs presumably recognize all serotype O strains since the five strains analyzed represent different phylogenetic clades. They bind the same antigenic site as M170F, which was previously shown to be a conserved site in serotypes A and O, and which has an altered 3D structure when 146S dissociates into 12S particles. M916F had the lowest limit of detection, which varied from 0.7 to 5.9 ng/mL 146S particles for three serotype O strains. Conclusions: We identified four VHHs (M907F, M910F, M912F, and M916F) that specifically bind 146S particles of probably all serotype O strains. They enable further improved FMDV vaccine quality control.

## 1. Introduction

Foot-and-mouth disease (FMD) is an infectious disease that affects cloven-hoofed animals. It causes large economic losses to the livestock industry, especially in developing countries. FMD is caused by the FMD virus (FMDV), an aphthovirus of the *Picornaviridae* family. The RNA genome of FMDV is surrounded by a protein capsid that consists of 60 protomers, each comprising one copy of the four capsid proteins VP1, VP2, VP3, and VP4. The latter protein is located on the capsid inner surface [1]. Five protomers assemble into a pentamer (12S), and twelve pentamers assemble into a complete FMDV capsid. There are seven FMDV serotypes: A, O, C, Asia1, SAT1, SAT2, and SAT3. The most prevalent serotypes are O and A [2]. FMD can be controlled effectively through vaccination [3,4]. However, due to antigenic variation, vaccines do not always elicit cross-protection even for strains within the same serotype. Conventional FMD vaccines are based on chemically inactivated FMDV strains formulated with an adjuvant. Furthermore, FMD virus-like particles (VLPs), which consist of empty capsids lacking the RNA genome, are produced with mammalian-, insect-, or bacterial-cell expression systems for use in vaccines [5,6].

FMDV virions sediment at 146S in sucrose density gradients (SDGs). When exposed to mild heating (56 °C) or an acidic pH (<6.5), the intact FMDV capsid dissociates into stable VP4-deficient pentameric 12S particles, resulting in a strongly reduced induction of neutralizing antibodies [7,8,9]. Such capsid dissociation can also occur during vaccine manufacturing, formulation, and storage [10,11,12]. The 146S content of vaccines should thus be monitored closely. The quantification of FMDV 146S antigens for use in vaccines is traditionally performed by SDG fractionation and 146S quantification by 254–260 nm UV absorbance to measure RNA content [13,14]. Double-antibody sandwich (DAS) ELISAs for FMDV antigen quantification were also developed using polyclonal [15,16,17] or monoclonal [18,19,20,21,22] antibodies (mAbs). These assays are more sensitive and scalable than the aforementioned methods but are generally not 146S-specific. Different DAS-ELISAs were previously developed that make use of 146S-specific mAbs to quantify the 146S content of serotype A and O strains [23,24]. In several studies, we used recombinant single-domain antibodies derived from camelid heavy-chain-only antibodies, known as VHHs [25], for the development of 146S-specific DAS-ELISAs. We initially discovered that a panel of twenty-four previously isolated VHHs against the FMDV strain O1/Manisa/TUR/69 [26] contained two 146S-specific VHHs, M170F [10] and M210F [27], and one 12S-specific VHH, M3F [9]. DAS-ELISA, employing M3F and M170F, proved to be highly useful in measuring the stability of the FMDV antigens before and after oil emulsification and showed stabilizing effects of the excipients [10]. Further 146S-specific VHHs were isolated against strains Asia1/Shamir/ISR/89 and SAT2/SAU/2/00 [28] and several serotype A strains [27]. Furthermore, a 12S-specific ultralong bovine antibody suitable for the analysis of FMD integrity was recently reported [29]. Generally, 146S-specific VHHs and mAbs are serotype-specific and highly strain-specific [23,24,27,28]. For example, M170F binds O1/Manisa/TUR/69 but not O/TAW/3/97, while the VHHs M678F and M688F bind only 2 out of 15 serotype A strains analyzed [26,27]. However, for serotype A, broadly strain-reactive 146S-specific VHHs were also previously identified. M691F was found to bind all 15 strains analyzed, while M702F bound 8 out of 15 serotype A strains analyzed [27]. Since many different strains are used in FMDV vaccines produced by different manufacturers [30], such VHHs are more valuable for use in vaccine quality control.

Similar to conventional antibodies, VHHs have three complementarity-determining regions (CDRs) that exhibit high amino acid sequence variation and constitute the antigen-binding domain, known as the paratope. The third CDR (CDR3) domain is the most variable in both sequence and length. Clones with similar CDR3 sequences generally originate from the same B-cell clone and have diverged due to somatic hypermutation [31,32]. After the isolation of a panel of antigen-binding VHHs from an immunized llama, the VHHs can often be allocated to clonal groups based on a comparison of CDR3 sequences. VHHs from the same clonal group generally bind the same antigenic site but often differ in antigen-binding affinity and specificity. Indeed, clonally related FMDV-binding VHHs often differ in their strain specificity [26,27], but always fall into the same epitope bin [26,33]. Instead of screening for natural diversity in VHH sequences, synthetic diversity can also be introduced in VHHs to improve antigen-binding properties. CDR3 often strongly contributes to antigen binding [31,32], as is also observed for M170 [34]. Phage-display selection from a CDR3-mutagenized secondary library previously yielded VHHs [35,36,37] or single-chain Fvs [38,39] with improved affinity [35,36,39], functional activity [37], or specificity [38].

In the present work, we aimed to isolate 146S-specific VHHs against FMDV serotype O that are more broadly reactive to serotype O strains. For this purpose, we carried out the following:Screened previously isolated VHHs M81F and M208F that are clonally related to proven 146S-specific VHHs M210F and M170F [26], respectively.Yeast-produced VHHs Nb45F and Nb205F that were reported to be 146S-specific [40].Generated mutant VHH M918F by CDR3 mutagenesis of M170F.Isolated novel VHHs M907F, M910F, M912F, M915F, and M916F from phage-display immune libraries generated previously.

Only the latter method was successful. It resulted in four novel 146S-specific VHHs, M907F, M910F, M912F, and M916F, that were broadly reactive with serotype O strains. Several antigen-binding properties of these novel VHHs that are relevant for their use in FMDV vaccine quality control were then determined.

## 2. Materials and Methods

### 2.1. VHHs and FMDV Antigens

VHHs that were used and yeast-produced are shown in Table 1.

The FMDV strains A/TUR/20/2006, A/MAU/1/2006, A/KEN/12/2005, A/TUR/14/98, A22/IRQ/24/64, A24/Cruzeiro/BRA/55, Asia1/Shamir/ISR/89, C1/Detmold/FRG/60, SAT2/EGY/2/2012, O1/BFS 1860/UK/67, O1/Manisa/TUR/69, O/TUR/5/2009, O/SKR/7/2010, and O/TAW/3/97 were produced as previously described [27]. Briefly, FMDV was amplified in BHK-21 cells grown in suspension in industrial-size bioreactors or 850 cm^2^ roller bottles. FMDV present in the clarified supernatant was inactivated with 10 mM binary ethylenimine (BEI) and concentrated using polyethylene glycol-6000 precipitation, resulting in crude antigen. The FMDV strain Asia1/BAR/8/2009 was produced in BHK-21 cells grown in 850 cm2 roller bottles without further BEI inactivation or PEG precipitation.

Purified 12S and 146S particles were obtained as described previously [27,28]. Briefly, FMDV particles were fractionated using 10–40% sucrose density gradients (SDGs) by centrifugation for 2 h at 10 °C and 200,000× *g*. After the collection of 20 separate fractions, the absorbance at 254 nm was determined to identify the 146S peak. The 146S concentration in µg/mL was then calculated by multiplying the absorbance at 254 nm by 126.7. The 12S particles were generated by heating crude antigen for 1 h at 56 °C. The concentration of 12S particles was derived from the 146S concentration of the sample from which it was prepared.

### 2.2. Generation of CDR3-Randomized M170 Phage-Display Library

A novel phage-display library with the randomization of eight CDR3 residues (GFALPPSD) of M170 was prepared based on an approach using a TRI nucleotide Mixture (TRIM) oligonucleotide, also known as trimer phosphoramidites, to avoid the introduction of unwanted amino acids such as cysteine, stop codons, and frame shifts, as depicted in Appendix A. TRIM oligonucleotides are synthesized using a mixture of trinucleotides for primer synthesis based on amino acid codons rather than individual nucleotides, enabling the precise control of amino acid diversity and codon representation at individual mutagenized positions. The TRIM oligonucleotide BOLI410 (5′-GCGGTTTACTATTGCACCGCG X01 X02 X03 X04 X05 X06 X07 X08 TATTGGGGCCAGGGTACC-3′, Ella Biotech GmbH, Fürstenfeldbruck, Germany) was randomized at codon positions X01 to X08 in CDR3 since this region encompasses many contact residues in the structural data of M170F complexed with O/BY/CHA/2010 [34]. These eight randomized positions have amino acid frequencies based on the natural llama antibody repertoire diversity, which were calculated based on the abYsis database [41], as represented in Appendix A using WebLogo (https://weblogo.berkeley.edu/logo.cgi; accessed on 25 February 2025). Furthermore, the existing amino acid of the M170 sequence at that specific location was kept at the highest frequency, along with arginine, threonine, serine, alanine, and tyrosine in slightly higher proportions, while methionine, proline, leucine, and isoleucine were kept in low proportions. The DNA sequence of M170 (EMBL database acc. no. AJ811551) cloned into plasmid pRL188 [42] was used as a template in a splice overlap extension PCR utilizing primers BOLI6 (5′-CCTTTTCCTTTTGGCTGGTTTTGC-3′), BOLI415 (5′-CCGCGGTGCAATAGTAAACCGCCGTGTCCTCAGGTTTCAG-3′), BOLI410, and BOLI412 (5′-CTAGCGGCCGCTGATGACACGGTAACTTGGGTACCCTG-3′), as described in Appendix A. The final amplicon was cloned into the pRL144 vector after digestion with restriction endonucleases PstI HF and NotI HF [28]. Transformation of electrocompetent Escherichia coli TG1 cells (Lucigen Corporation, Middleton, WI, USA.) resulted in a library of 3 × 10^7^ unique clones.

### 2.3. Phage-Display Selection of VHHs

Phage-display selections of 146S-specific VHHs were carried out using the CDR3-randomized M170 phage library and libraries from immunized llamas described previously. Llama 9245 was immunized with SDG-purified 146S particles of O1/Manisa/TUR/69 [27] while llama 6058 was immunized with the crude antigen of O1/Manisa/TUR/69 [26]. Llamas 7211 and 7212 were immunized by natural infection with FMDV strains O1/BFS 1860/UK/67 and O1/Manisa/TUR/69, respectively [26]. Parallel biopannings were performed with all libraries, using SDG-purified 146S particles of O/SKR/7/2010, which were captured with either VHH M8F or M31F. Competition with soluble 10 μg/mL A22/IRQ/24/64 12S particles was performed during biopanning on O/SKR/7/2010 146S captured with M31F, but not using M8F, since M31F is highly specific for serotype O FMDV [26] while M8F binds to A22/IRQ/24/64 12S particles [27]. Phage-display selections were performed by two consecutive rounds of biopanning in polystyrene 96-well plates as described previously [27,28]. Briefly, plates were coated with low concentrations of VHH (0.1–1 μg/mL) for the subsequent capture of FMDV 146S particles (1 μg/mL) in PBS containing 1% milk and 0.05% Tween-20 (PBSTM). Plates were then subsequently incubated with phages displaying VHHs in PBSTM. Bound phages were finally eluted by incubation with 1 mg/mL trypsin in PBS for 30 min at 37 °C and immediately transduced to *Escherichia coli* TG1 [(F′ traD36 proAB lacIqZ ΔM15) supE thi-1 Δ(lac-proAB) Δ(mcrB-hsdSM)5(rK− mK−)] cells. To monitor the progress of phage-display selection, a parallel phage ELISA was carried out (details in Section 2.5 below).

After the second round of panning, phages were transduced to *E. coli* TG1 cells, and individual colonies were picked. The production of soluble recombinant VHHs, directed to the periplasm, was induced using 1 mM isopropyl β-d-thiogalactopyranoside. They were then analyzed by ELISAs as described in Section 2.5 below. Positive clones were sequenced as described previously [27,28]. Numbering the VHH residues and defining the CDR and framework (FR) regions were carried out according to the IMGT system [43]. VHHs were allocated to clonal groups based on identical CDR3 length and at least 70% CDR3 amino acid sequence identity.

### 2.4. Yeast Production of VHHs

The dimeric VHH M3ggsVI-4_Q6E_ was described previously [28]. The M170 VHH coding region [26] was cloned in plasmid pRL507 [42] for the production of M170J in yeast. All further VHHs were produced using plasmid pRL188 and yeast strain SU51 as described previously [42]. The latter VHHs are referred to by the suffix F. They partially dimerize through disulfide bond formation using a cysteine residue in the hinge region. This cysteine residue is mutated to serine in pRL507, resulting in the production of strictly monomeric VHHs in the case of M170J. Due to using the *Bst*EII restriction endonuclease site for cloning purposes, which is located in FR4, IMGT residue 127 of M81 is mutated from Ala to Ser.

VHHs were analyzed by reducing SDS-PAGE, using precast 4–12% gradient gels (Thermo Fisher Scientific, Rockford, IL, USA), and stained using GelCode Blue reagent (Thermo Fisher Scientific). VHHs were digested with endoglycosidase H (New England Biolabs, Ipswich, MA, USA) according to the manufacturer’s instructions.

### 2.5. ELISAs

Various ELISA procedures were used. We first describe the basic ELISA procedure that was used in all procedures. ELISAs were performed by coating high-binding polystyrene 96-well plates (Greiner, Solingen, Germany) with 0.1–1 μg/mL of unlabeled VHH in 50 mM carbonate/bicarbonate buffer, at a pH of 9.6 (coating buffer), overnight at 4 °C. Coating and subsequent incubations were performed using 100 µL per well. After washing, the coated plates were incubated with the FMDV antigen in the ELISA buffer (1% skimmed milk; 0.05% Tween-20; 0.5 M NaCl; 2.7 mM KCl; 2.8 mM KH_2_PO_4_; 8.1 mM Na_2_HPO_4_; pH 7.4) or PBSTM for 45 min at RT. Plates were subsequently incubated with either phage-displayed VHH, *E. coli*-produced VHH, or yeast-produced biotinylated VHH, and subsequently with a suitable specific horse radish peroxidase (HRP) conjugate, using the same buffer as used for incubation with FMDV antigens (ELISA buffer or PBSTM). After staining with 3,3′,5,5′-tetramethylbenzidine (TMB), the reaction was stopped by the addition of 0.5 M H_2_SO_4_ (50 µL per well), and the absorbance at 450 nm (A450) was measured using a Spectramax ABS Plus spectrophotometer (Molecular Devices, Sunnyvale, CA, USA).

#### 2.5.1. ELISA Using *E. coli*-Produced Phage-Displayed VHHs

For phage ELISA, plates were coated with VHHs and subsequently with FMDV antigens as described in Section 2.3. However, trypsin elution was replaced by incubation with HRP-conjugated anti-M13 g8p antibodies (Antibody Design Labs, San Diego, CA, USA) in PBSTM, followed by TMB staining.

#### 2.5.2. ELISA Using E. coli-Produced Soluble VHHs

To evaluate the 146S specificity of *E. coli*-produced soluble VHHs obtained by phage-display selection, nine ELISAs with different capturing VHH/FMDV particle combinations were conducted, with two parallel comparisons carried out in each ELISA. The two comparisons used the M8F- or M31F-captured O/SKR/7/2010 146S versus 12S particles, with an additional fifth ELISA to monitor binding to M3F-captured O/SKR/7/2010 12S particles. Further comparisons used M170F and M3F for capturing O1/Manisa/TUR/69 146S and 12S particles, respectively, and M23F and M3F for capturing O/Taiwan/3/97 146S and 12S particles, respectively. For each of these nine ELISAs, plates were coated with 1 μg/mL yeast-produced VHH and then subsequently incubated with 4 μg/mL FMDV particles in PBSTM. Bound FMDV particles were detected by incubation with 10-fold-diluted *E. coli*-produced soluble VHH, which contains a myc-tag, and an HRP-conjugated mAb against the c-myc tag (clone 9E10; Roche Applied Science, Mannheim, Germany), followed by TMB staining.

#### 2.5.3. ELISA Analysis of SDG Fractions Using Yeast-Produced VHHs

DAS-ELISAs employing various yeast-produced VHHs were used to determine FMDV particle concentrations in SDG fractions as follows: Plates were coated with 0.5 μg/mL of unlabeled VHH and subsequently incubated with serial 2-fold dilution series of FMDV particle preparations in ELISA buffer. Standards of FMDV particles were included in the ELISAs for the quantification of FMDV particles in SDG fractions. Standards were serially diluted 2-fold using both 1 μg/mL and 0.66 μg/mL as starting concentrations. Only for the M3F ELISAs were the SDG fraction samples and standards heated for 1 h at 56 °C prior to ELISA. Plates were subsequently incubated with 0.25 μg/mL of a biotinylated VHH. The different DAS-ELISAs employed the same VHH for coating in an unlabeled form and for detection in the biotinylated form. Bound biotinylated VHHs were detected with 1 μg/mL HRP-conjugated streptavidin (Jackson ImmunoResearch Laboratories Inc., West Grove, PA, USA). Absorbance data were evaluated using an Excel spreadsheet template (Microsoft Corporation, Redmond, WA, USA). A four-parameter logistic curve was fitted to absorbance and FMDV particle concentrations of standards by non-linear least squares using the Excel solver tool. The FMDV particle concentration in SDG fractions was then determined by interpolation.

#### 2.5.4. ELISA Analysis of Particle Specificity of Yeast-Produced VHHs

To assess the binding of VHHs to particular FMDV particles, the particles were titrated in two- or three-fold dilution series in a similar DAS-ELISA as described in the previous section. The effective concentration (EC) of particles required to reach an absorbance value of 1 was then interpolated after four-parameter logistic curve fitting of absorbance and particle concentrations. The limit of detection (LOD) of ELISAs was measured by titrating untreated and heated FMDV particles in triplicate and interpolating the FMDV particle concentration required to reach the average absorbance value of the background plus 3 times the standard deviation. Along with the above titrations, untreated FMDV antigen samples, which consist predominantly of 146S, were also titrated in ELISA buffer containing 4 μg/mL of heated FMDV antigen (12S) of the same strain to analyze the interference of 12S particles with the quantification of 146S particles.

#### 2.5.5. Heterologous DAS-ELISA

A (heterologous) DAS-ELISA was also carried out with a matrix of different combinations of unlabeled and biotinylated yeast-produced VHHs, using both untreated and heated FMDV antigens at 1 μg/mL without titration.

#### 2.5.6. Mapping of Antigenic Sites by Blocking/Competition ELISA

The ability of VHHs to bind independent antigenic sites was studied by blocking/competition DAS-ELISA. Initially, the optimal concentration of biotinylated VHH for competition was determined by the titration of biotinylated VHH without competition, as described above for determining EC values. A biotinylated VHH concentration was selected that provided about 80% of the maximal absorbance value observed with the highest VHH concentration analyzed. ELISAs were performed by coating unlabeled VHH (0.5 µg/mL) for the subsequent capture of crude FMDV antigens that contain 1 µg/mL 146S in addition to about 20% 12S particles. Plates containing VHH-captured FMDV were first incubated with the unlabeled VHH (5 µg/mL) in 90 µL/well for 30 min (blocking step). Then, without washing plates, 10 µL biotinylated VHH in the predetermined concentration was added and incubated for another 30 min (competition step). The same VHH as used for coating was also used as biotinylated VHH. A control without antigen and a control without biotinylated VHH were included. Bound biotinylated VHH was detected by incubation with HRP-conjugated streptavidin and TMB staining. The % inhibition of antigen binding due to a competing VHH was calculated as 100 − 100 × ([A450 with competing VHH] − [A450 without antigen coating])/([A450 without competing VHH − A450 without antigen coating]).

### 2.6. Affinity Measurements by Biolayer Interferometry

The Octet Red96 System (Sartorius, Fremont, CA, USA) was used for affinity measurement based on biolayer interferometry. An assay temperature of 30 °C was used. PBS containing 0.05% Tween-20 (PBST) was initially used as the kinetics buffer. High-precision streptavidin (SAX) sensors (Sartorius) were loaded with biotinylated M170F or M210F, each at 2 µg/mL, for 300 s, and then with O1/Manisa/TUR/69 or O/SKR/7/2010 146S particles, respectively, each at 2 µg/mL in PBST for 900 s. Sensors were subsequently incubated in PBST containing 10 mM DTT (PBSTD) for 900 s (baseline step). Then, the association of serial dilutions of unlabeled VHHs in PBSTD was carried out for 300 s, and finally, dissociation occurred for 300 s. For some VHHs, assays were also carried out without 10 mM DTT using PBST for VHH association and dissociation phases. Correction for baseline drift occurred using a reference sensor without unlabeled VHH. The on-rate (*k_a_*) and off-rate (*k_d_*) were determined by global fitting of the association and dissociation phases of a series of unlabeled VHH concentrations, assuming a 1:1 stoichiometry of VHH to FMDV binding. The equilibrium dissociation constant (*K_D_*), a measure of affinity, was then calculated as the ratio of *k_d_* to *k_a_*. Octet Analysis Studio v12.2 software (Sartorius) was used for data analysis.

### 2.7. FMDV Phylogenetic Analysis

Different pools of FMDV strains circulate in different geographic regions. Within these pools, different genetic lineages of FMDV (topotypes) can be differentiated based on the sequencing of the VP1, which is the most variable capsid protein. Initially, 8 topotypes were identified for serotype O that were named Europe–South America (Euro-SA), Middle East–South Asia (ME-SA), Southeast Asia (SEA), Cathay, West Africa (WA), East Africa (EA), Indonesia-1 (ISA-1), and Indonesia-2 (ISA-2) [44]. Later, the EA topotype was split into 4 topotypes named EA-1, EA-2, EA-3, and EA-4 [45]. Within some topotypes, distinct (sub)lineages can be differentiated. The ME-SA topotype comprises the lineages Ind-2001, PanAsia, PanAsia2, and non-lineage strains such as O1/Manisa/TUR/69. The SEA/Mya-98 lineage encompasses several important vaccine strains.

We aimed to perform phylogenetic analysis of the serotype O vaccine strains used in this study and at least one representative of each topotype. We used the P1 region encoding the 4 capsid proteins for phylogenetic analysis, rather than only the VP1 coding region, since the major 146S-specific antigenic site is primarily located on VP3 [9]. A large phylogenetic tree of serotype O P1 sequences is given by Reeve et al. 2016 [45]. For strain O1/Manisa/TUR/69, we used a sequence obtained in our laboratory (FN594747). For the other vaccine strains used in this study, we searched the GenBank database based on strain name. For O/SKR/7/2010, such a P1 sequence could not be found. We used the sequence of O/SKR/6/2010 instead. The Food and Agriculture Organization website provides prototype strains and database accession numbers for the various topotypes/lineages [46]. These often only comprise VP1. In such cases, we used BLAST searches (https://blast.ncbi.nlm.nih.gov/blast.cgi; accessed on 25 and 29 October 2024) to find full P1 sequences showing at least 99% identity in VP1 to the prototype sequence. In this manner, P1 DNA sequences could be retrieved for all topotypes except ISA-2. The sequence of strain O/UGA/17/98 (EU919245) contained only 2185 bp of the P1 region due to a 3′ truncation. All further P1 sequences comprised 2202 to 2205 bp.

Phylogenetic analysis was carried out using the MegAlign Pro program of the Lasergene version 17 suite (DNASTAR, Madison, WI, USA). The P1 DNA sequences were aligned using MAFFT. A maximum likelihood phylogenetic tree was generated with 1000 cycles of bootstrap sampling.

### 2.8. Molecular Visualization

Pymol 2.5.2 (Schrödinger, LCC, New York, NY, USA) was used for the molecular visualization of VHH FMDV complexes [47].

## 3. Results

### 3.1. Generation and Selection of Novel 146S-Specific VHHs with Broader Strain Recognition

M210F and M170F are 146S-specific VHHs that recognize strain O1/Manisa/TUR/69 but not O/TAW/3/97 [26,27]. M170F also does not recognize strain O/BY/CHA/2010 [27], which is highly related to O/SKR/7/2010 (phylogenetic analysis in Section 3.3). Several approaches were employed to obtain VHHs that show 146S-specific binding to all serotype O FMDV strains. We screened the literature for potential 146S-specific VHHs with published sequences (Figure 1a). Two VHHs, Nb45 and Nb205, were isolated against O/BY/CHA/2010 and claimed to specifically recognize 146S particles [40]. We previously observed that clonally related VHHs for serotype A FMDV often retain their 146S-specific binding but may differ in strain recognition [27]. Therefore, M81 and M208, which are clonally related to M210F and M170F, respectively [26], were also selected for further analysis. Although the VHH sequences of Nb45, Nb205, M81, and M208 were published, they were not yeast-produced and thoroughly characterized earlier. Since we have currently analyzed the 146S specificity and strain recognition of these yeast-produced VHHs, we consider them novel VHHs for the purpose of this work. VHH names are color-coded to identify clonal groups of VHHs (Table 1).

Further VHHs were obtained by biopanning using a serotype O-specific VHH for the capture of O/SKR/7/2010 146S particles, which enables competition with an excess of serotype A 12S particles during phage-display selection to counterselect against VHHs binding 12S particles, which is similar to earlier published procedures [27,28]. We used phage-display libraries from llamas previously immunized with O1/Manisa/TUR/69 or O1/BFS 1860/UK/67. Since different FMDV strains were used for immunization and biopanning, the resulting VHHs should recognize at least two serotype O strains. We also generated a next-generation phage-display library of M170 with the randomization of eight residues in CDR3 that are known contact sites for binding to FMDV [34], as shown in Figure 1b. CDR3 randomization was performed using a TRIM oligonucleotide approach that is described in Appendix A. This randomized M170 library was similarly used in biopanning to isolate novel VHHs specific to 146S particles of O/SKR/7/2010. Biopanning from immune libraries yielded five novel VHHs, M907, M910, M912, M915, and M916, that comprise four clonal groups based on CDR3 sequence homology (Table 1). M915 is part of the same clonal groups as M170 and M208; they differ by at most three amino acids (Figure 1a). The further four VHHs form three novel clonal groups; M910 and M912 are part of the same clonal group since their CDR3 of nineteen residues differs by only four residues (Table 1). Biopanning of the M170 CDR3-randomized library yielded M918, which has two mutations compared to M170. Mutation L110Y is located within the randomized region of CDR3, whereas mutation N85K is located in the middle of FR3 (Figure 1a). Mutation L110Y is located in the middle of the CDR3 region where M170F makes close contact with O/BY/CHA/2010 in the cryogenic electron microscopy (cryo-EM) structure, whereas N85K is more distant (Figure 1b). Mutation N85K is not in the (CDR3) region selected for randomization. This mutation probably occurred as an artifact during PCR amplification used for second-generation library construction. This suggests that L110Y was primarily selected by biopanning using O/SKR/7/2010 146S.

Thus, we obtained ten novel VHHs that could have broader FMDV serotype O strain recognition combined with 146S-specific binding. Four VHHs were published, one VHH was generated by CDR3 mutagenesis of a published VHH, and five VHHs were selected from phage-display immune libraries generated previously. We underlined the VHH names in the figures and tables to differentiate previously characterized control VHHs (underlined) from these ten novel characterized VHHs (not underlined).

### 3.2. Yeast Production of VHHs

The novel isolated VHHs were produced in yeast to assess their 146S and strain specificity. Yeast production was achieved using plasmid pRL188 that encodes a C-terminal extension with the natural llama heavy-chain antibody long hinge, including a single cysteine residue, which allows partial disulfide bond formation and thus VHH dimerization [48]. VHHs produced in this manner are indicated by the suffix F. To assess whether such dimerization affected antigen binding, control VHH M170 was also produced as a strictly monomeric VHH by mutation of the cysteine residue located in the hinge region to serine, using plasmid pRL507 [42], resulting in M170J. Reducing SDS-PAGE analysis shows an about 22 kDa band representing intact VHH (Appendix A), which is slightly diffuse due to O-glycosylation in the hinge region [48], in the case of most VHHs. However, for Nb45F and Nb205F, such a band was absent, and a smear in the low-molecular-weight region probably represents the presence of degraded VHH (Appendix A). These VHHs could thus not be produced in yeast. VHH degradation was also observed for M910F, although most VHHs were still intact. A sample of M170F that was analyzed without DTT contains an additional band at about 40 kDa (hashtag sign) that represents disulfide-bonded VHH dimers. As expected, this 40 kDa band is absent in M170J (Appendix A).

VHHs M208F, M915F, and M916F contain a potential N-glycosylation site in the same location at IMGT position 57 (Table 1). However, this site was not recognized for all three VHHs since they did not contain an additional band migrating more slowly than the about 22 kDa band, and treatment with endoglycosidase H, which removes N-glycosyl groups, did not affect their migration (Appendix A). Therefore, these VHHs were further used in this study without endoglycosidase H treatment. Part of the yeast-produced purified VHHs was biotinylated for use in DAS-ELISAs as previously described [42].

### 3.3. FMDV Strain and Particle Specificity of VHHs

The specificity of VHHs for 146S or 12S particles, as well as the specificity for FMDV strains, was analyzed by DAS-ELISA. Most antigens were BEI-inactivated and used at 1 µg/mL 146S or 12S. However, a crude culture supernatant of FMDV Asia1/BAR/8/2009-infected BHK21 cells was used without BEI inactivation and unknown antigen concentration since the BEI-inactivated antigen was not available. The ELISA with the non-inactivated Asia1/BAR/8/2009 antigen was performed within a high-containment unit, while the other ELISAs were performed separately in a conventional laboratory. The ELISAs with M918F, Nb45F, and Nb205F resulted in a relatively high background absorbance, which was especially high with the ELISA carried out in the high-containment unit (Figure 2). The control VHHs M3F, M8F, and M220F were bound to many FMDV strains of different serotypes. These three ELISAs also resulted in high absorbance values with the Asia1/BAR/8/2009 antigen, indicating that it contained sufficient antigens for detection in VHH-based DAS ELISAs.

We further discuss the strain and particle specificity of individual VHHs displayed in Figure 2. The VHHs that were degraded on SDS-PAGE, Nb45F and Nb205F, were negative in ELISA with each antigen, confirming that they were completely degraded. M332F and M377F showed 146S-specific binding to serotype Asia 1 and SAT2, respectively, as previously described [27,28]. However, M332F did not recognize strain Asia1/BAR/8/2009, which again supports the notion that 146S-specific VHHs are highly strain-specific. M81F and M210F of clonal group red recognized four serotype O strains, but not to O/TAW/3/97. The novel VHH M81F showed lower 146S specificity than M210F. M170F, M208F, M915F, and M918F of clonal group light blue recognized all three serotype O strains, but not O/TAW/3/97. Only M208F recognized O/SKR/7/2010, although with a relatively low absorbance value (1.076). The three further VHHs of the light blue clonal group did not bind O/SKR/7/2010. This is surprising since M915F and M918F were selected by phage display using O/SKR/7/2010. This indicates that these VHHs recognized O/SKR/7/2010 with reduced affinity as compared to O1/Manisa/TUR/69 and required multivalent binding, such as that occurring during phage display for binding O/SKR/7/2010. Four VHHs, M907F, M910F, M912F, and M916F, showed 146S-specific binding to all five serotype O strains analyzed. Absorbance values were relatively low for M910F, which is most likely related to its partial degradation (Appendix A). They comprise three clonal groups: dark red, gray, and purple. They were obtained from llamas previously immunized with O1/Manisa/TUR/69 by phage-display selection on O/SKR/7/2010. This suggests that a cross-selection on a different strain helped improve strain recognition. All VHHs that bound serotype O strains in a 146S-specific manner did not bind FMDV strains from the other four serotypes studied.

Phylogenetic analysis was performed for the five serotype O strains used in this study for determining antigenicity, together with a diverse set of FMDV serotype O strains that should represent the serotype O genetic lineages (topotypes) observed worldwide in diverse geographic regions. The classification into 11 topotypes and further sublineages [44,45], as well as the rationale for the selection of strains and the P1 region for performing phylogenetic analysis, is described in Materials and Methods. We obtained P1 regions of all 11 topotypes, except the ISA-2 topotype. The five strains used in this study cover four topotypes that are phylogenetically quite distinct (Figure 3). This implies that VHHs binding to all five strains analyzed will likely also cross-react with further serotype O strains.

### 3.4. FMDV Particle Specificity of VHHs

The FMDV particle specificity of VHHs (Figure 4) was determined using a method similar to that previously outlined [27,28]. It relies on the ratio of the effective concentration (EC) of 12S compared to 146S, resulting in an absorbance value of 0.5.

A 12S/146S EC ratio (Table 2) above 10 is considered to be 146S-specific. We thus titrated both untreated (146S) and heated (12S) antigens of FMDV strains O1/Manisa/TUR/69, O/SKR/7/2010, and O/TAW/3/97 in ELISA utilizing the novel serotype O VHHs (Figure 4). As expected, the 12S-specific control VHH M3F showed preferential binding of 12S, while the control VHH M8F bound to 12S and 146S for all three strains. The 12S/146S EC ratio (Table 2) of M3F most likely reflects the fraction of 12S present in the untreated antigen. M170F shows only a slightly lower EC value (29 ng/mL) than M170J (43 ng/mL) using O1/Manisa/TUR/69 146S, showing that bivalent binding due to intermolecular disulfide bond formation is not necessary. M81F shows a low 146S specificity for strain O1/Manisa/TUR/69 (EC ratio = 2.2), whereas the related VHH M210F shows a much higher 146S specificity (EC ratio above 41). The four VHHs, M907F, M910F, M912F, and M916F, that showed 146S-specific binding to all five serotype O strains analyzed have different EC values using 146S, which is also reflected in the EC ratio, since none of them bound to 12S at the highest concentration analyzed. For all three FMDV strains analyzed, the partially degraded M910F shows the lowest EC ratio, while M916F consistently shows the highest EC ratio (>117 to >208).

We subsequently assessed the particle specificity of the ten serotype O-specific VHHs by analysis of FMDV particles fractionated by SDGs for three serotype O strains (Figure 5). The samples subjected to SDG fractionation contained 12S particles since half of the sample was heated prior to SDG fractionation. As a result, a 12S and a 146S peak could be identified by M3F or M8F ELISAs. The 12S peak is less sharp for O/TAW/3/97 as compared to O1/Manisa/TUR/69 and O/SKR/7/2010, and the FMDV antigen is also detected in fractions 6 to 12, probably due to the aggregation of O/TAW/3/97 12S particles. All ten serotype O-specific VHHs showed high specificity for 146S particles fractionated by SDGs. Only M81F ELISA detected O1/Manisa/TUR/69 12S particles, although at low levels, requiring an extended secondary axis for fractions 1–10 to clearly visualize such binding (Figure 5).

The particle specificity of VHHs in DAS-ELISAs was also determined using different combinations of VHHs for coating and as biotinylated VHH (Appendix A). We used seven serotype O-specific VHHs with high 146S specificity, representing all five clonal groups, in combination with three control VHHs, M3F, M8F, and M23F, that recognize different antigenic sites [26], as well as a VHH multimer, M3ggsVI-4_Q6E_, that contains the M3 VHH linked to a porcine immunoglobulin binding VHH [28]. The source of 146S was crude antigen that contains at least 20% 12S. As a result, the ELISA signals observed with M3F/M3ggsVI-4_Q6E_ combined with M8F or M23F on untreated antigen can be explained by the presence of 12S. Many combinations of 12S-specific VHHs such as M3F or M3ggsVI-4_Q6E_ with one of the seven 146S-specific VHHs resulted in absorbance values above those of the background. These absorbance values were especially high with coated M210F, coated M916F, biotinylated M170F, and biotinylated M210F, combined with either M3F or M3ggsVI-4_Q6E_ (Appendix A). This contradicts the high particle specificity observed in Figure 2, Figure 3 and Figure 4 using DAS-ELISAs employing the same VHH for coating and detection. This indicates that 146S specificity of VHHs is not absolute and relies on using particle-specific VHHs both during coating and as biotinylated VHHs for optimal particle specificity in DAS-ELISAs.

### 3.5. Epitope Mapping of VHHs by Competition ELISAs

Four independent antigenic sites, indicated by roman numerals I to IV, recognized by 24 VHHs against O1/Manisa/TUR/69 were previously identified by epitope binning [26]. Similar epitope binning was performed for nine serotype O 146S-specific VHHs that represent all clonal groups and four control VHHs, M3F, M8F, M23F, and M220F, defining these four antigenic sites (Figure 6). M81F, of the same clonal group as M210F and M170J, which binds the same site as M170F, was excluded from this analysis. All 146S-specific VHHs recognized the same antigenic site that we previously defined as site I/III for M170F [26]. Most 146S-specific VHHs showed non-reciprocal competition with M8F that defines site I. Such non-reciprocal competition of M8F was previously observed with 146S-specific VHHs against Asia1/Shamir/ISR/89 and A/TUR/14/98 [26]. Furthermore, non-reciprocal competition was observed with M910F and M912F of the gray clonal group in combination with the seven other 146S-specific VHHs. This probably reflects a low binding affinity of these VHHs, which is also suggested by their high EC values for strain O1/Manisa/TUR/69 (Table 2). Thus, all serotype O 146S-specific VHHs recognize the same antigenic site or a partially overlapping antigenic site.

### 3.6. VHH Binding Affinity for FMDV

We determined the affinity of seven novel serotype O 146S-specific VHHs and the control VHHs M210F and M170F for O1/Manisa/TUR/69 and O/SKR/7/2010 using biolayer interferometry. M910F, Nb45F, and Nb205F were excluded because of their low binding in the ELISA (Figure 4). A 1:1 interaction model was used for curve fitting of VHH association and dissociation phases of sensorgrams (Appendix A) to determine the *k_a_*, *k_d_*, and *K_D_* values (Table 3). The affinity of M208F or M916F binding to O/SKR/7/2010 and M912F binding to O1/Manisa/TUR/69 could not be determined (Table 3) since binding was not detected in biolayer interferometry (Appendix A). This is most likely due to DTT treatment, which was necessary for the reduction of the intermolecular disulfide bonds between the C-terminal cysteines that cause partial dimerization and avidity. For M916F, we measured affinity for O1/Manisa/TUR/69 using both buffers with (reduced) and without DTT (oxidized). Reduced M916F had a low affinity (*K_D_* = 264 nM) due to a fast dissociation (high *k_d_*), while oxidized M916F had a much slower dissociation. Due to the high baseline drift observed with several concentrations of oxidized M916F (Appendix A), the *K_D_* of 3.1 nM was only based on a single curve fit and thus unreliable (R^2^ = 0.84; Table 3). DTT treatment thus resulted in about a tenfold decrease in apparent affinity (avidity) of M916F. A similar comparison of reduced and oxidized M170F binding to O1/Manisa/TUR/69 did not, however, show a reduced apparent affinity due to DTT treatment. Consistent with this, M170J, which lacks the C-terminal cysteine required for multimerization, exhibits a similar affinity to M170F (Table 3). Thus, DTT treatment reduces the affinity of M916F but not M170F. The further seven 146S-specific VHHs show high-affinity binding to O1/Manisa/TUR/69 with *K_D_* values ranging from 1.7 to 29 nM. However, binding to O/SKR/7/2010 was only observed for three (reduced) VHHs. They showed a lower affinity compared to O1/Manisa/TUR/69, with *K_D_* values ranging from 4.4 to 104 nM (Table 3).

### 3.7. Limited Validation of 146S-Specific ELISAs

M907F and M916F ELISAs were further validated for their application in the quantification of 146S particles based on the highly specific and sensitive detection of 146S particles of all five serotype O strains analyzed in ELISA. Untreated (146S) and heated (12S) FMDV antigens of three strains were titrated in ELISA in triplicate for the determination of the limit of detection (LOD). The 146S particles were also titrated in the presence of a constant high concentration of 12S particles to determine interference with 146S detection. The 12S concentration (4 µg/mL) chosen for analyzing interference was also used in an earlier similar analysis [28], and above the maximal concentration of 2 µg/mL of 146S that was used in ELISAs in this study. Both VHHs showed a high specificity for 146S particles for all three strains analyzed, utilizing untreated antigen in the absence and presence of 12S. However, 12S particles were not detected even at the highest concentration of 2 µg/mL (Figure 7). The LOD for 146S particles ranged from 7 to 22 ng/mL for M907F and from 0.7 to 5.9 ng/mL for M916F (Table 4). M916F showed a lower LOD than M907F with all three strains and, most notably, with O/TAW/3/97.

## 4. Discussion

### 4.1. Isolation of 146S-Specific VHHs

We previously isolated ten 146S-specific VHHs for FMDV serotype A that included two VHHs, M691F and M702F, that showed broad recognition of serotype A strains, while eight further VHHs showed high strain specificity [27]. Such high strain specificity was also observed with two previously isolated 146S-specific VHHs for serotype O, namely, M170F and M210F [26]. In this study, we used several approaches to isolate novel 146S-specific VHHs that broadly recognize serotype O strains. We could not produce two VHHs, Nb45 and Nb205, that were previously claimed to be FMDV serotype O 146S-specific [40] in an intact form in baker’s yeast, preventing their further analysis. However, all further VHHs were successfully produced in yeast. Nb45F and Nb205F are both camel VHHs, while all further VHHs originated from llamas. These camel VHHs both have an additional disulfide bridge between a cysteine at IMGT position 35 and a cysteine in CDR3 (Figure 1a), which is typically observed with camel VHHs [50,51]. This suggests that the additional disulfide bond of these camel VHHs could have caused their degradation upon yeast production. However, we are surprised that this occurs with both VHHs since we and previous researchers successfully produced many VHHs with an additional disulfide bond in yeast after their isolation from *E. coli* by phage display. Thus, we are concerned that the quality of the reported sequences of Nb45 and Nb205, which were isolated from a single non-immunized camel [40], could have caused the inability to reproduce them by yeast expression. This could be checked by the precise reproduction of these VHHs in *E. coli* as published earlier.

The successfully yeast-produced VHHs included ten serotype O-specific VHHs showing high 146S specificity, which were obtained by different methods. These ten VHHs consist of two previously published VHHs, M170F and M210F [26], with known 146S specificity [27], and two further VHHs, M208F and M81F [26], which were selected based on their clonal relationship (similar CDR3 sequence) with M170F and M210F, respectively. A fifth VHH, M915F, which was also clonally related to M170F, and four further VHHs (M907F, M910F, M912F, and M916F) that comprise three novel clonal groups were isolated by phage-display selection on O/SKR/7/2010 146S particles from immune libraries that were generated after llama immunization with O1/Manisa/TUR/69. The tenth VHH, M918F, was isolated by the CDR3 randomization of M170F.

None of the three novel VHHs of the M170F and M210F clonal groups showed substantially broadened strain recognition; only M208F was bound to one additional strain, O/SKR/7/2010, although at lower absorbance values. Attempts to broaden the strain recognition of M170F by the randomization of CDR3 were also not successful. This may partly be due to the relatively small secondary library size (3 × 10^7^ clones), which is insufficient to capture the full diversity resulting from the simultaneous mutagenesis of eight residues, many of which contain mutant residues represented at frequencies below 5% (Appendix A). A more effective strategy, as previously suggested [36], involves first identifying residues tolerant to mutation via alanine scanning, followed by focused mutagenesis at a reduced number of positions to ensure complete coverage in the secondary libraries. Additionally, our approach did not include the mutagenesis of other CDRs, even though CDR1 substantially contributes to M170F antigen binding (Figure 1b). However, four 146S-specific VHHs, M907F, M910F, M912F, and M916F, recognized all five serotype O strains analyzed. This shows that VHHs with high 146S specificity and broad strain recognition are not abundantly present and need careful selection by phage display, preferably using different strains for immunization and phage-display selection, as we previously observed for serotype A, such as Asia1 and SAT2 146S-specific VHHs [27,28]. The use of phage-display selection in combination with camelid VHHs, which naturally results in a large functional library size [25,52], likely played a significant role in isolating these rare VHHs. In contrast, conventional mAbs with such uncommon specificities are less frequently identified, as they typically rely on less effective selection systems, such as hybridoma screening or phage-display selection from immune libraries with compromised functionality due to the reshuffling of VL and VH domains.

### 4.2. Immunological Characterization of 146S-Specific VHHs

M907F, M910F, M912F, and M916F presumably recognize all serotype O strains since phylogenetic analysis showed that the five FMDV strains used represent four FMDV topotypes. The choice of FMDV strains for analyzing antigenicity was determined by the availability of well-validated antigen batches that were mostly obtained from the vaccine production facility in Lelystad, which was formerly part of Wageningen University. As a result, we could not analyze binding to all FMDV topotypes. The four topotypes with confirmed binding to M907F, M910F, M912F, and M916F occur primarily in Europe, the Middle East, Asia, and South America. Future experimental studies should be conducted to confirm the presumed binding of these four VHHs to all eleven topotypes, and especially for the four East African topotypes and the single West African topotype, since these form a separate phylogenetic clade (Figure 3) and FMDV is still endemic in Africa [4]. M907F, M910F, M912F, and M916F represent three novel clonal groups as compared to the previously isolated 146S-specific VHHs with higher strain specificity. This is consistent with previous findings where serotype A 146S-specific VHHs that were broadly strain-reactive represented two separate clonal groups as compared to the more strain-specific VHHs [27]. All novel serotype O 146S-specific VHHs were found to recognize the same antigenic site that overlaps with the M170F site. The M170 site was mapped mainly on VP3, close to the border where two pentamers interact, using cryo-EM analysis of a complex of M170 with O/BY/CHA/2010 [34]. This site overlaps with site 4 defined by mAb escape mutants [9,49]. Interestingly, cross-linking mass spectrometry analysis showed that this site was also recognized by two VHHs, M691F and M702F, which showed broad recognition of serotype A strains [33]. The serotype A 146S-specific VHHs recognize two independent antigenic sites, one of which is recognized by M691F, M702F, and M678F, while the other site is recognized by M688F [33]. Cryo-EM studies showed that M678F was bound to a similar position as M170, while M688F was bound to a different site located on VP1 that is distant from the pentamer border [9]. Thus, for both serotypes A and O, VHHs that are both broadly strain-reactive and 146S-specific recognize the same antigenic site.

The 146S specificity of a VHH, or antibody, is not a binary parameter that can be expressed as specific or not specific, but varies to different degrees between different combinations of FMDV strains and VHHs [27,28]. A comparison of the cryo-EM structures of intact capsids and 12S particles showed that the 146S specificity of M170 [34] and M688F [9] is due to an altered 3D structure of the epitopes when capsids dissociate, whereas the 146S-specific binding of M678F relies on binding an epitope crossing the pentamer border that is disrupted upon dissociation into 12S particles [9]. The 12S-specific binding of M3F, however, relies on binding an epitope that is internal in the capsid and only accessible for antibodies after capsid disruption [9]. The variation in 146S specificity of VHHs is probably related to the dependence on the difference in the 3D structure of the epitope between 12S and 146S particles [9,34]. Therefore, similar to our previous studies [27,28], we quantified the 146S specificity of VHHs specific for serotype O by the calculation of the ratio between EC values obtained with 12S and 146S particles (Table 2). Two VHHs that represent different clonal groups, M907F and M916F, showed a high 146S specificity for the three serotype O strains analyzed, based on this EC ratio. This high 146S specificity was confirmed by the analysis of SDG fractions (Figure 4). The particle specificity of VHHs was further analyzed in heterologous DAS-ELISAs using different VHHs for coating and as biotinylated detection reagents. Remarkably, combinations of the 12S-specific VHH M3, either as an M3F monomer or M3ggsVI-4_Q6E_ dimer, with several 146S-specific VHHs, resulted in absorbance values above those of the background (Appendix A). Contrary to the gradual nature of 146S specificity, the 12S specificity of M3 is more absolute since its epitope is completely interior in 146S particles [9]. As a result, the absorbance values obtained in DAS-ELISAs using M3 combined with a 146S-specific VHH more likely indicate a low-affinity cross-reaction of the 146S-specific VHH with 12S particles. We previously observed cross-linking of the 146S-specific VHH M702F with three residues at the two-fold symmetry axis that are completely buried in 146S particles and only surface-accessible after dissociation into 12S particles, and suggested that this indicates particle flexibility at the two-fold symmetry axis [33]. However, it may also represent a low-affinity cross-reaction with 12S particles. These assays also suggest that the high 146S specificity observed with some VHHs in DAS-ELISAs is dependent on using the same VHH for both coating and biotinylated VHHs.

In addition to a high 146S specificity, M907F and M916F also showed the sensitive detection of 146S particles, which was not affected by the presence of a high amount of 12S particles. M916F showed a lower LOD than M907F for the three strains analyzed (Table 4) and also appeared to recognize these strains with more sensitivity than most other VHHs in a single titration series (Figure 4). This is contrary to the low affinity of M916F observed for both O1/Manisa/TUR/69 (*K_D_* = 264 nM) and O/SKR/7/2010, where binding could not be detected utilizing DTT-reduced VHHs by biolayer interferometry. VHHs were reduced since they were produced using an expression format that results in partial dimerization through a C-terminal cysteine present in the llama heavy-chain antibody long-hinge region [48], which is expected to increase binding avidity [53]. However, VHHs also have a canonical disulfide bond, which is internal in the VHH 3D structure, and which stabilizes VHH and is essential for the antigen binding of most VHHs. Therefore, VHH reduction was performed using a low DTT concentration (10 mM) to reduce the intermolecular disulfide bond of the solvent-accessible C-terminal cysteine, without reducing the intramolecular canonical disulfide bond. A similar mild reduction of intermolecular disulfide bonds without reducing the canonical disulfide bond was previously found to require the optimization of the DTT concentration used for each individual VHH [54]. Indeed, reduced M916F appeared to have a much lower affinity using a non-optimized DTT concentration of 10 mM. We also analyzed the effect of disulfide bond reduction using M170F, both by using reduced and non-reduced M170F as well as using a separate expression format lacking the C-terminal cysteine, M170J, in affinity measurements (Table 3) and ELISAs (Figure 2 and Figure 4). This did not show a reduced binding of monomeric M170 as compared to its partially disulfide-bonded counterpart, and affinity remained unaltered for M170F and M170J upon VHH reduction. Thus, it appears that 10 mM DTT caused a reduction of the canonical disulfide bond in M916F but not in M170F.

### 4.3. Application of Novel 146S-Specific VHHs

FMD VLPs are currently also being developed for use as vaccines [5,6]. They are not infectious since they lack the RNA genome. This offers the advantage of their production in facilities requiring lower biocontainment levels, but the disadvantage is the quantification by the traditional SDG method, which requires the detection of the RNA genome by UV absorbance [13,14], which is not possible. As a result, ELISAs employing capsid-specific VHHs that do not bind 12S particles show great promise for VLP quantification. We have previously demonstrated for both FMDV serotypes A and O that not all 146S-specific VHHs possess the capability to recognize VLPs [27]. Hence, there is still a need to ascertain whether the four novel 146S-specific broadly strain-reactive VHHs M907F, M910F, M912F, and M916F also recognize VLPs.

However, these VHHs, and especially M907F and M916F, show great promise for the quantification of 146S particles of all serotype O strains. Their high specificity for 146S particles is an important improvement as compared to a previously described ELISA for the quantification of FMDV serotype O 146S particles [22] that also detects 12S particles. Size-exclusion high-performance liquid chromatography (SE-HPLC) is nowadays also used for the quantification of FMDV antigens [12,55,56,57,58,59]. As compared to SE-HPLC and the traditional SDG method for the quantification of FMDV antigens [13,14], ELISA-based methods offer the advantage of higher sensitivity and higher sample throughput. A universal ELISA for the detection of intact FMDV particles of all serotypes for application in vaccine quality control was recently published [60]. The ELISA employs an mAb against a conserved epitope near the N-terminus of the VP4 capsid protein that requires mild glutaraldehyde treatment of capsids to improve antibody binding. The applicability of SE-HPLC, SDGs, and the VP4-ELISA for use with strains from different serotypes offers advantages for their broad applicability with monovalent vaccines composed of many different strains, but it also complicates the analysis of multivalent vaccines composed of strains from different serotypes. However, the serotype specificity that is observed with 146S-specific VHHs enables the analysis of multivalent vaccines.

## 5. Conclusions

In this study, we used several approaches to obtain novel 146S-specific VHHs that broadly recognize serotype O strains. Mutagenesis of earlier isolated strain- and 146S-specific VHHs, M170F and M210F, either by the CDR3 randomization of M170F or by the characterization of earlier identified VHHs that are clonally related to M170F or M210F, did not yield VHHs with the desired specificities. Only after careful novel phage-display selection from immune libraries could we isolate four 146S-specific VHHs that recognize all five serotype O strains analyzed. These VHHs, M907F, M910F, M912F, and M916F, represent three novel clonal groups. They all bound the same antigenic site as M170F, which was previously mapped by cryo-EM. They showed a high specificity for 146S particles in SDGs. M907F and M916F showed the highest sensitivity for most FMDV strains. In further validation ELISAs, M907F and M916F showed an LOD ranging from 0.7 to 22 ng/mL that was not affected by the presence of 4000 ng/mL 12S particles. Phylogenetic analyses suggest that these VHHs recognize all eleven serotype O topotypes, which needs experimental confirmation. Furthermore, the recognition of VLPs still needs to be determined. Nevertheless, these novel VHHs show promise for application in ELISAs to monitor the quality of serotype O FMDV in monovalent and multivalent vaccines.

## Figures and Tables

**Figure 1 vaccines-13-00500-f001:**
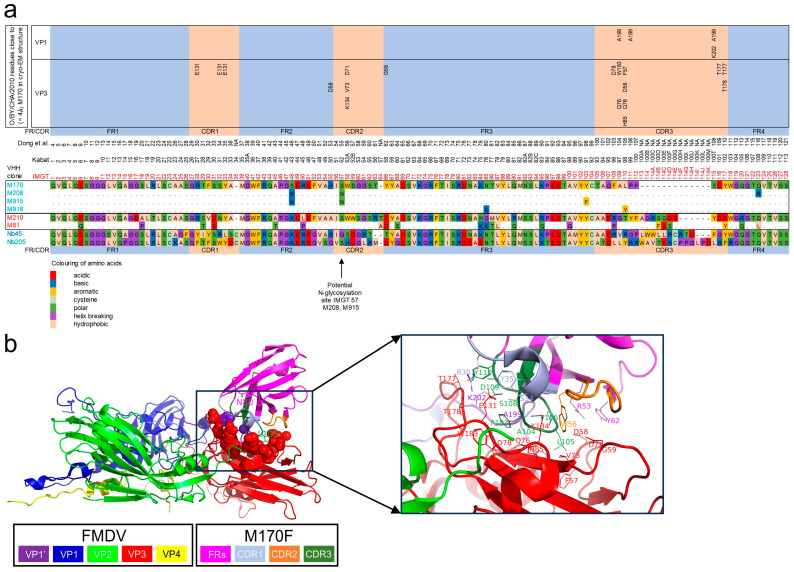
Structure of published VHHs or clonally related to published VHHs. (**a**) The mature VHH protein sequences of 8 VHHs as produced in yeast. The three VHHs clonally related to M170 are shown below the M170 sequence, where dots represent their identity in relation to M170. M81, which is clonally related to M210, is similarly shown below the M210 sequence. The IMGT system [43] was used for alignment and numbering of VHHs as well as defining the different complementarity-determining regions (CDRs) and framework regions (FRs). The Kabat numbering scheme is given for reference. Amino acids were classified and color-coded based on their physicochemical properties. The VP1 and VP3 residues within 4 Å distance of a M170 residue based on the cryo-EM structure of M170 complexed to O/BY/CHA/2010 are indicated [34]. The M170 residue numbering used by Dong et al. [34] is given for reference as well. (**b**) Cartoon presentation of M170 complexed to O/BY/CHA/2010 (PDB: 7DST). VP1, VP2, VP3, and VP4 of a single protomer are shown in addition to the VP1 C-terminal of 20 amino acids of an adjacent protomer (VP1′), which is close to the M170 binding site. The left panel shows the side chains of the two M170F residues mutated in M918F as sticks and the side chains of the VP1/VP3 contact sites as spheres. This region is enlarged at the right, where side chains of FMDV and M170 contact sites are shown as sticks. Panel b was rendered using Pymol 2.5.2.

**Figure 2 vaccines-13-00500-f002:**
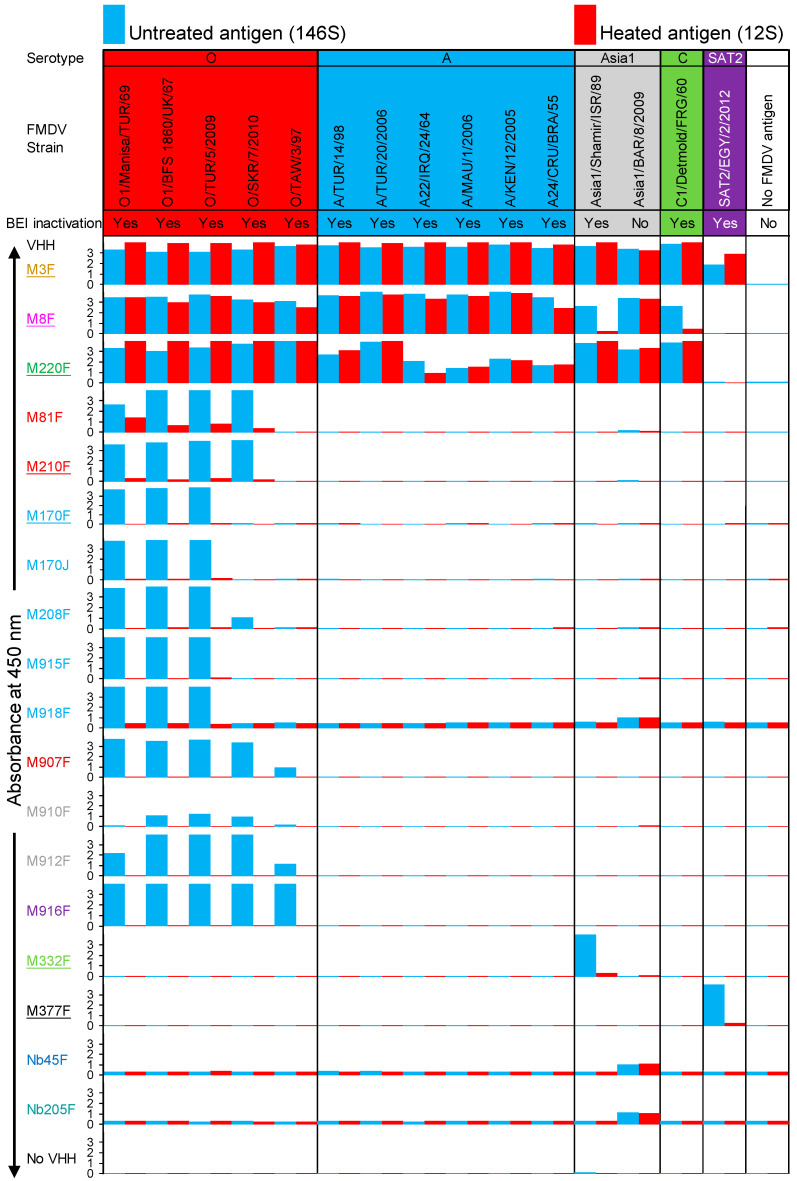
FMDV particle and strain specificities of VHHs. DAS-ELISAs were performed using a matrix of 18 VHHs and 15 FMDV strains representing 5 serotypes. Antigens from each strain were used in ELISA either untreated (146S; blue) or after conversion into 12S by heating for 1 h at 56 °C (red). Most antigens were BEI-inactivated and used at 1 µg/mL 146S. For the strain Asia1/BAR/8/2009, the crude culture supernatant of FMDV-infected BHK21 cells was used without BEI inactivation at a 5-fold dilution with unknown antigen concentration. VHHs are color-coded by their clonal group. Control VHHs isolated and characterized previously are underlined. The VHHs are arranged from top to bottom according to their clonal group and their particle and strain specificity. The ELISA with the strain Asia1/BAR/8/2009 that was not BEI-inactivated was performed separately within the high-containment unit of WBVR and resulted in a higher background absorbance value. This is most evident with M918F, Nb45F, and Nb205F, which already have a high background absorbance.

**Figure 3 vaccines-13-00500-f003:**
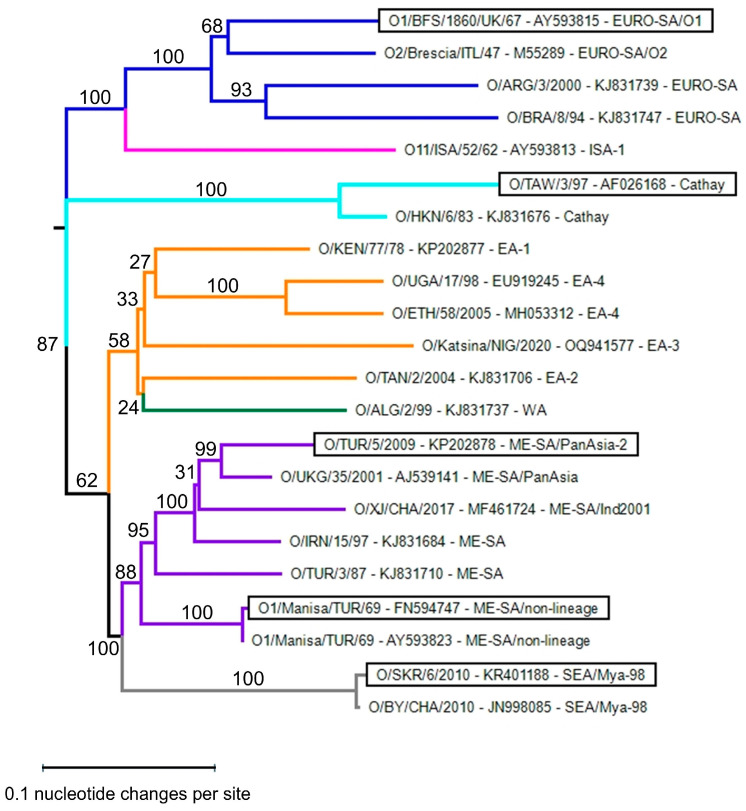
Sequence divergence of FMDV serotype O capsid encoding P1 region. A phylogenetic tree of P1 nucleotide sequences was generated using the maximum likelihood method and 1000 cycles for bootstrap sampling. Bootstrap values are indicated next to branches. For each sequence, the FMDV strain name, database accession number, and topotype/lineage are indicated and separated by dashes. Strains used to analyze antigenicity are boxed. Branches are color-coded according to their topotype, using orange for all 4 East African (EA) topotypes.

**Figure 4 vaccines-13-00500-f004:**
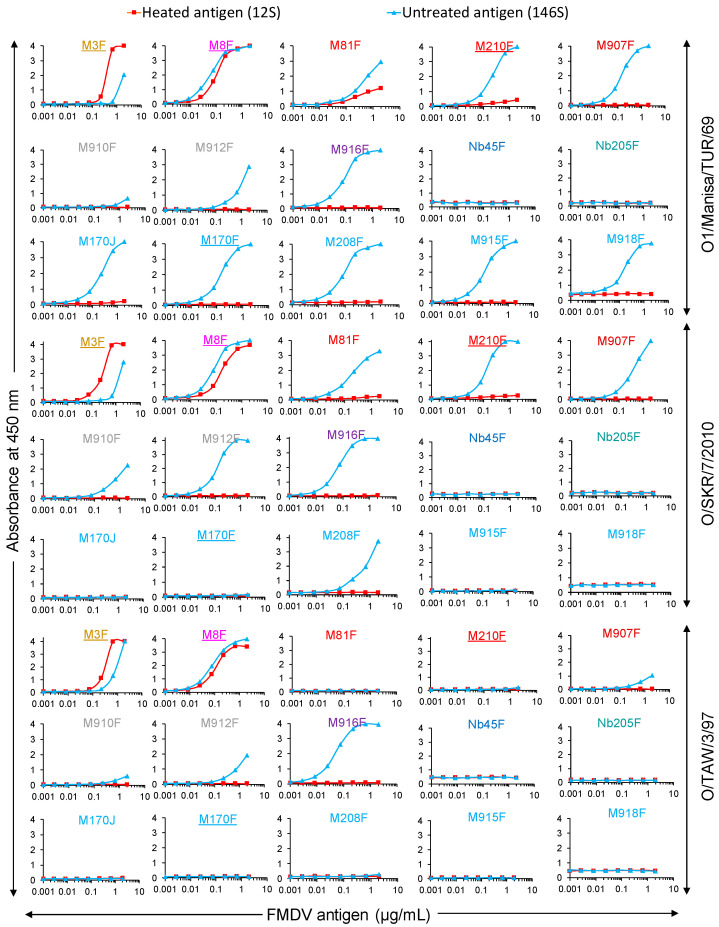
FMDV particle specificity of VHHs in DAS-ELISA. FMDV antigens of three serotype O strains, as indicated on the right, were titrated in DAS-ELISAs using various VHHs either as untreated FMDV antigen (146S) or after heating at 56 °C for 1 h (12S). VHHs are color-coded by their clonal group. Control VHHs isolated and characterized previously are underlined.

**Figure 5 vaccines-13-00500-f005:**
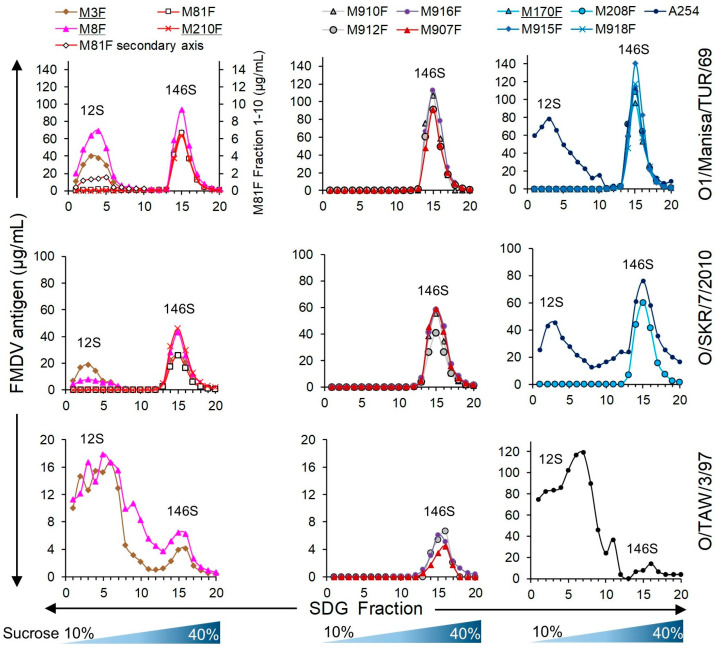
ELISA analysis of VHH specificity for FMDV particles fractionated by SDGs. BEI-inactivated antigens from the three FMDV strains indicated on the right were used for SDG fractionation. SDGs were layered with a 1:1 mixture of untreated and heated FMDV antigens to ensure the presence of both 146S and 12S. SDG fractions were analyzed by the measurement of absorbance at 254 nm to identify the 146S peak and by DAS-ELISAs using the broadly reactive VHHs M3F and M8F to identify all peaks. A further 10 ELISAs were performed using different 146S-specific VHHs. The names of control VHHs isolated and characterized previously are underlined. The M81F ELISA of O1/Manisa/TUR/69 fractions 1–10 is also shown on an extended secondary axis to visualize low levels of 12S particles that were detected.

**Figure 6 vaccines-13-00500-f006:**
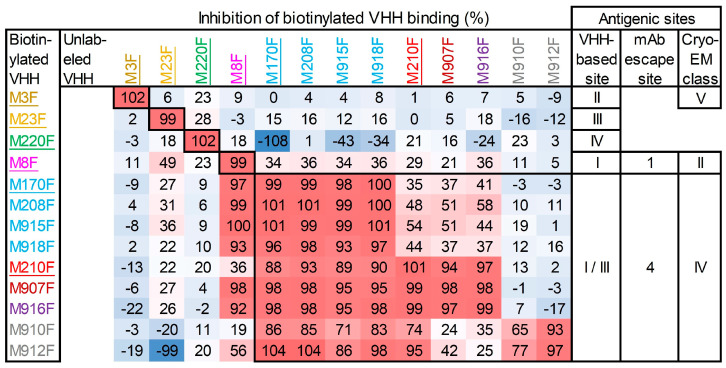
Epitope binning of VHHs by competition ELISAs using FMDV strain O1/Manisa/TUR/69. A red–blue coloring is used to visualize differences in percentage inhibition. The VHHs are color-coded by their clonal group. The control VHHs isolated previously are underlined. Antigenic sites are indicated by boxing. Antigenic sites identified previously by competition ELISAs using VHHs [26], isolation of mAb escape mutants [49], or cryo-EM analysis of antibody–FMDV complexes [9,34] are indicated on the right.

**Figure 7 vaccines-13-00500-f007:**
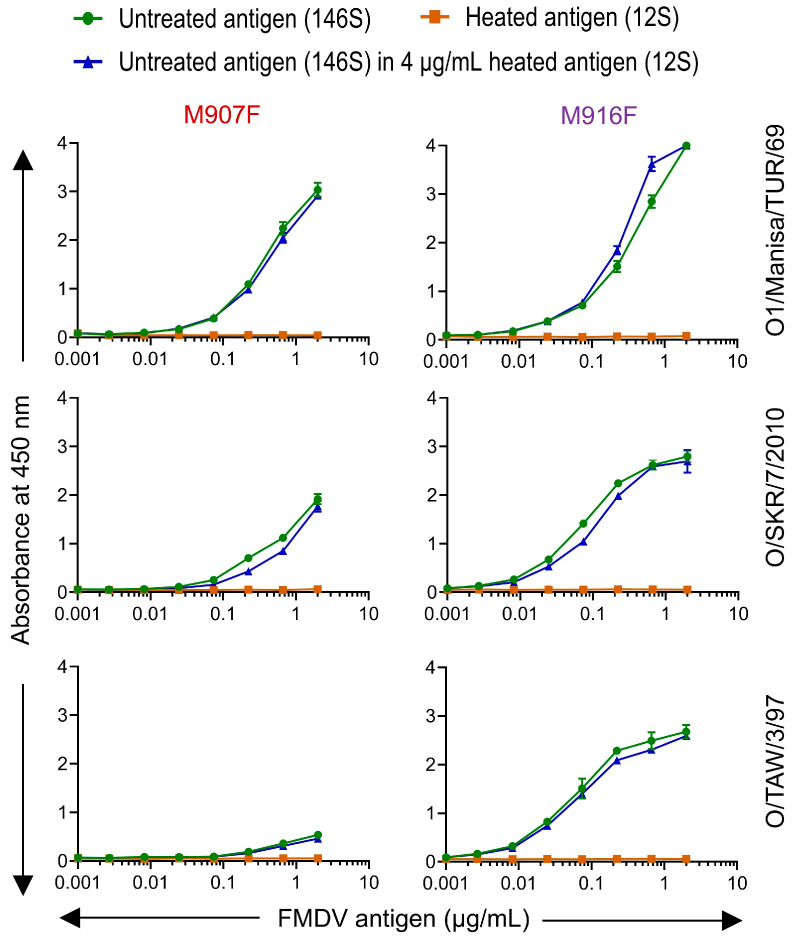
Capsid specificity and interference of 12S particles in capsid quantification in DAS-ELISAs using M907F or M916F. Untreated antigens consisting predominantly of 146S and 12S particles generated by heating three FMDV strains, as indicated on the right, were titrated in DAS-ELISAs with M907F or M916F, as indicated on top. Another titration was performed with a 3-fold dilution series of untreated FMDV particles to which a constant amount of 4 µg/mL 12S particles was added to measure the interference of 12S particles. The mean and standard deviation of triplicate measurements are presented.

**Table 1 vaccines-13-00500-t001:** Sequence characteristics and origins of VHHs used in this study.

VHH ^a^	Animal ^b^	CDR3 Sequence	VHH Subfamily ^c^	N-Glycosylation Site ^d^	Reference ^e^
Nb45F	Camel	AAERVRGPLWWLLRCRTDFGY	4	None	[40]
Nb205F	Camel	ATDLLYRRWAVTERCPPGLPDLRF	X	None	[40]
M3F	6058	AGDRSLTVVASSWRY	1	None	[26]
M3ggsVI-4_Q6E_	6058	AGDRSLTVVASSWRY	1	None	[28]
M8F	6058	NGLRASNAGWEPRFGT	2	None	[26]
M23F	6058	NLWSQWQSETY	2	None	[26]
M31F	6058	AAGLSIYSNTYYYTRGEEYTY	1	None	[26]
M81F	6666	AAEPGTYFAGRFESEYDY	1	None	[26]
M210F	6666	AAERGTYFAGRSQDEYDD	1	None	[26]
M220F	7211	AAGYRIDTQPMDRDFYYY	1	None	[26]
M332F	3049	AAAWSFRSDYGARLKSAYDF	1	None	[28]
M377F	3050	NALVLSSSWSEGDY	2	None	[28]
M170F	7212	TAGFALPPSDY	1	None	[26]
M170J	7212	TAGFALPPSDY	1	None	This study
M208F	7212	TAGFALPPSDY	1	57	[26]
M915F	7212	TAGFALPPSDY	1	57	This study
M918F	7212	TAGFAYPPSDY	1	None	This study
M907F	7211	AADNYHRSRYSGNYDYTDSWFGS	1	None	This study
M910F	7212	ARGAEAAGLGSHREYDYSY	1	None	This study
M912F	7212	ARGAEAAGWGSHHQYDYAY	1	None	This study
M916F	7212	AAEKSLILGTAVSGYDY	1	57	This study

^a^ VHHs are color-coded by their clonal group. VHHs isolated and characterized previously are underlined. ^b^ Numbers refer to individual llamas. ^c^ VHH subfamilies were defined previously [27,28]. ^d^ The IMGT position of the Asn residue of potential N-glycosylation sites is indicated. ^e^ Reference describing the original isolation or preparation of VHH. Nb45F, Nb205F, M81F, and M208F were yeast-produced and characterized only in this study and thus considered novel VHHs.

**Table 2 vaccines-13-00500-t002:** Specificity of VHHs for 12S and 146S particles in DAS-ELISA using the same VHHs for coating and as biotinylated VHHs.

	O1/Manisa/TUR/69			O/SKR/7/2010			O/TAW/3/97		
	EC_0.5_ (ng/mL)		EC	EC_0.5_ (ng/mL)		EC	EC_0.5_ (ng/mL)		EC
VHH ^a^	146S	12S	Ratio ^b^	146S	12S	Ratio	146S	12S	Ratio
M3F	1008	215	0.21	687	115	0.17	348	175	0.50
M8F	11	28	2.5	18	42	2.3	14	29	2.0
M81F	74	159	2.2	31	>2000 ^c^	>66	>2000	>2000	ND ^d^
M210F	49	>2000	>41	37	>2000	>53	>2000	>2000	ND
M170J	43	>2000	>47	>2000	>2000	ND	>2000	>2000	ND
M170F	29	>2000	>70	>2000	>2000	ND	>2000	>2000	ND
M208F	15	>2000	>133	70	>2000	>29	>2000	>2000	ND
M915F	21	>2000	>94	>2000	>2000	ND	>2000	>2000	ND
M918F ^e^	53	>2000	>38	>2000	>2000	ND	>2000	>2000	ND
M907F	32	>2000	>63	70	>2000	>29	578	>2000	>3.5
M910F	1611	>2000	>1.2	177	>2000	>11	1452	>2000	>1.4
M912F	223	>2000	>9	29	>2000	>69	300	>2000	>6.7
M916F	17	>2000	>117	14	>2000	>142	10	>2000	>208

^a^ VHHs are color-coded by their clonal group. VHHs isolated and characterized previously are underlined. ^b^ The EC ratio is the EC value of 12S divided by the EC value of 146S. ^c^ A value >2000 indicates absorbance does not exceed 0.5 at the highest VHH concentration analyzed. ^d^ ND, not determined. ^e^ EC values were calculated at A450 = 1 to compensate for the high background absorbance in M918F ELISA.

**Table 3 vaccines-13-00500-t003:** Binding affinity of VHHs for FMDV strains O1/Manisa/TUR/69 and O/SKR/7/2010 determined by biolayer interferometry.

VHH ^a^	FMDV Strain	*K_D_* (nM)	*k_a_* × 10^5^ (1/Ms)	*k_d_* × 10^−4^ (1/s)	R^2^
M81F	O1/Manisa/TUR/69	6.7	9.3	62	0.981
M81F	O/SKR/7/2010	104	6.4	669	0.874
M210F	O1/Manisa/TUR/69	1.7	24	40	0.983
M210F	O/SKR/7/2010	4.4	25	113	0.980
M170J without DTT ^b^	O1/Manisa/TUR/69	11	4.0	44	0.964
M170J	O1/Manisa/TUR/69	17	3.4	57	0.975
M170F without DTT	O1/Manisa/TUR/69	8.5	3.4	29	0.991
M170F	O1/Manisa/TUR/69	12	3.0	36	0.987
M208F	O1/Manisa/TUR/69	13	5.5	70	0.976
M208F	O/SKR/7/2010	ND ^c^			
M915F	O1/Manisa/TUR/69	29	1.7	49	0.988
M918F	O1/Manisa/TUR/69	18	1.3	24	0.984
M907F	O1/Manisa/TUR/69	7.9	1.6	13	0.997
M907F	O/SKR/7/2010	59	1.7	98	0.978
M912F	O1/Manisa/TUR/69	ND			
M916F	O1/Manisa/TUR/69	264	1.9	512	0.995
M916F	O/SKR/7/2010	ND			
M916F without DTT	O1/Manisa/TUR/69	3.1	5.8	18	0.842

^a^ The VHHs are color-coded by their clonal group. Control VHHs isolated and characterized previously are underlined. ^b^ VHH sample in PBST without DTT. All further VHHs are in PBST with 10 mM DTT. ^c^ ND, insufficient VHH binding to determine affinity (Appendix A).

**Table 4 vaccines-13-00500-t004:** Limit of detection of M907F and M916F ELISAs for three FMDV strains.

FMDV Strain	LOD ^a^ (µg/mL)					
	M907F			M916F		
	146S	146S + 12S ^b^	12S	146S	146S + 12S ^b^	12S
O1/Manisa/TUR/69	0.015	0.018	>2.0 ^c^	0.0059	0.0025	>2.0
O/SKR/7/2010	0.009	0.007	>2.0	0.0010	0.0010	>2.0
O/TAW/3/97	0.022	0.017	>2.0	0.0007	0.0017	>2.0

^a^ The LOD was calculated based on the average absorbance value of the background plus three times the standard deviation of the data in Figure 7. ^b^ 146S particle titration in buffer containing 4 µg/mL 12S particles. ^c^ LOD not reached at the highest 12S particle concentration used (2 µg/mL).

## Data Availability

Full sequences of previously published VHHs are shown in Figure 1a. Only CDR3 sequences of novel isolated VHHs are disclosed. VHH proteins can be obtained from the authors at prices covering their production costs. All other data are included in the manuscript.

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
