# Peer review of "Single-Domain Antibodies That Specifically Recognize Intact Capsids of Multiple Foot-and-Mouth Disease Serotype O Strains"

_vaccines, 2025, doi:10.3390/vaccines13050500_

Round 1
Reviewer 1 Report
Comments and Suggestions for Authors
The authors have done a great job and obtained very interesting results.
It is difficult to understand throughout the manuscript which antibodies were previously obtained and which were created in this work. The manuscript often gives the quantity of antibodies, but lacks an identifier, which makes it difficult to understand. For example, "We identified four VHHs" (lane 23), antibodies should be listed here. Please review and edit this throughout the manuscript.
Some comments below:
Lines 36-37. Please add references.
Lines 56-58. What is the advantage of using camel antibodies for 146S-specific DAS-ELISAs over monoclonal antibodies? Why do monoclonal antibodies not provide the necessary specificity to recognise 146S? And why are camel antibodies more specific?
Lane 329-340. Are the antibodies described here taken from the literature? It is not clear. Please clarify. Why were these antibodies chosen?
Line 342. "procedures to counter select against VHHs binding 12S particles". Please explain.
Lane 346. Please provide details on the construction of the residue randomisation library.
Lane 350. How were the conjugate groups determined?
Lane 393. "These VHHs could thus not be produced in yeast". Please specify which antibodies were produced in yeast. Why did you not use another expression system for antibodies that cannot be produced in yeast?
Lane 541. Please specify how antigenic sites I-IV were selected. What is their sequence and length?
Please include a "Conclusion" section. It should state the main conclusions. Which antibodies have potential for DAS-ELISA? What ensures efficient binding of 146S particles?
Reviewer 2 Report
Comments and Suggestions for Authors
The authors performed important research on isolation of 146S-specific single-domain antibodies that recognize all serotype O strains.
The background of research is well written, previously published materials are well analyzed by the authors methods are clearly described.
Screening of previously isolated VHHs, the CDR3 randomized M170 phage library and libraries from immunized llamas were used. ELISA procedures were formulated. Novel isolated VHHs were produced in yeast to assess their 146S- and strain-specificity. DAS-ELISAs were performed using a matrix of 18 VHHs and 15 FMDV strains representing 5 serotypes, in the Figure FMDV particle specificity of VHHs in DAS-ELISA is clearly presented.
Validation of M907F and M916F ELISAs was performed for quantification of 146S particles.
The research is logically organized.
High 146S specificity was revealed, M907F and M916F demonstrated sensitive detection of 146S particles.
As a result, considered VHHs showed great promise for quantification of 146S particles of all serotype O strains. Their high specificity for 146S particles represents great benefit, and 146S-specific VHHs enables analysis of multivalent veterinary vaccines. To sum up, the research solves a relevant problem.
The manuscript can be published after minor revision.
The conclusion part to the article would be desirable.
Ref. 20. Article number 002032
Ref. 26 doi: 10.1016/j.jim.2004.12.005
Ref. 28 please exclude e1377
Ref. 41 Article number 1764
Reviewer 3 Report
Comments and Suggestions for Authors
This study makes a highly relevant and meaningful contribution to FMD research by isolating and characterizing four novel single - domain antibodies (VHHs) that specifically recognize intact 146S particles of multiple serotype O FMDV strains. These cross-reactive 146S-specific VHHs serve as valuable tools for monitoring capsid integrity across diverse FMDV strains, addressing a critical challenge in FMDV vaccine quality control. Although the study is methodologically rigorous and well-structured, there are several questions that need to be addressed to enhance the validity and applicability of the findings.
- Line 354-355, “Biopanning of the M170 CDR3-randomized library yielded M918, which has two mutations compared to M170”. The CDR3 mutagenesis approach yielded only M918F, which showed limited improvement in cross-reactivity. However, the manuscript does not discuss why this strategy failed to generate more diverse or effective variants. Was the library size (3×10⁷ clones) sufficient? A deeper analysis of the limitations of this method (e.g., insufficient diversity) is needed.
- Line 459-461, “We obtained P1 regions of all 11 topotypes, except the ISA-2 topotype. The five strains used in this study cover 4 topotypes that are phylogenetically quite distinct (Figure 3)”. The five tested serotype O strains (O1/Manisa/TUR/69, O/SKR/7/2010, etc.) represent four topotypes (ME-SA, SEA, Euro-SA, Cathay). However, strains of topotypes like ISA-2 and EA-1/2/3/4 were not used to analyze antigenicity. This raises questions about the universal applicability of the VHHs. The authors should clarify their rationale for strain selection and address potential biases. Are there any relevant research plans in the future?
- Line 594-596, “The 146S particles were also titrated in the presence of a constant amount (4 µg/mL) of 12S particles to determine interference with 146S detection”. Why was a constant amount of 4 µg/mL of 12S particles chosen for this titration? Were there any specific considerations or previous studies that influenced this concentration selection?
- Line 614, 4. Discussion. It is recommended to streamline the discussion section appropriately so that readers can focus more easily on the key points of this study.
